# Discriminative Self-Supervised Pre-Training for Esophagitis Detection in Upper GI Endoscopy Images

**Tobias Friedetzki**[1] iD                                       TOBIAS.FRIEDETZKI@UNI-WUERZBURG.DE
**Naveen Chandraiah**[1]                                   NAVEEN.CHANDRAIAH@STUD-MAIL.UNI-WUERZBURG.DE
**Emil Svoboda**[2] iD                                                    SVOBODA@UFAL.MFF.CUNI.CZ
**Pavel Pecina**[2] iD                                                       PECINA@UFAL.MFF.CUNI.CZ
**Frank Puppe**[1]                                                     FRANK.PUPPE@UNI-WUERZBURG.DE
**Adrian Krenzer**[1] iD                                           ADRIAN.KRENZER@UNI-WUERZBURG.DE

[1] *Julius-Maximilians University of Würzburg, Würzburg, Germany*

[2] *Charles University, Prague, Czech Republic*

**Editors:** Accepted for publication at MIDL 2026

## Abstract

Early and accurate detection of esophagitis in upper gastrointestinal endoscopy is essential for guiding targeted treatment and preventing progression to severe diseases such as esophageal cancer. Although deep learning methods have shown promise in supporting esophagitis diagnosis, their performance heavily relies on large amounts of labeled data, which are scarce. Consequently, supervised models often struggle to generalize to the high visual variability and subtle lesion differences encountered in real-world endoscopic examinations. In this work, we study discriminative self-supervised pre-training as a means of leveraging large-scale unlabeled data for robust representation learning. Multiple Vision Transformer models are pre-trained using the DINO framework on 395,201 unlabeled gastrointestinal endoscopy images and subsequently fine-tuned on a curated esophagitis dataset from three clinical centers. Our results demonstrate that self-supervised pre-training on in-domain endoscopic images significantly improves esophagitis detection performance compared to supervised pre-training on natural image datasets such as ImageNet. Specifically, in-domain DINO pre-training yields an average performance gain of 6.60 percentage points in AUPRC on the downstream detection task, with the best-performing model achieving an AUPRC of 89.82%. These findings highlight the importance of in-domain self-supervised learning for reducing annotation dependency and improving model robustness in upper GI endoscopy analysis.

**Keywords:** Esophagitis, Deep Learning, DINO, Upper GI Endoscopy, Self-Supervised

## 1. Introduction

Esophagitis is an inflammatory condition of the esophageal mucosa and represents a common finding in clinical gastroenterology. It can arise from multiple etiologies, including gastroesophageal reflux disease (GERD), medication-induced mucosal injury, infectious causes such as Candida, and immune-mediated disorders such as eosinophilic esophagitis (EoE) (Vakil et al., 2006; Zografos et al., 2009; Muir and Falk, 2021). In the context of chronic GERD, persistent inflammation may lead to metaplastic changes of the esophageal epithelium, known as Barrett's esophagus, which substantially increases the risk of esophageal adenocarcinoma (Souza, 2016). Early and reliable detection of esophagitis is therefore essential to guide therapy and prevent long-term disease progression.

In routine clinical practice, esophagitis is primarily diagnosed during upper gastrointestinal (GI) endoscopy. To enhance detection and support gastroenterologists during diagnosis, deep learning methods could be employed. Although such approaches have shown promise in esophagitis detection, their performance strongly depends on large amounts of labeled data, which remain scarce in this clinical domain.

Recent advances in self-supervised learning have demonstrated remarkable success in natural language processing, most notably through large language models pretrained on massive unlabeled corpora (Achiam et al., 2023; Touvron et al., 2023). Similar progress in computer vision has shown that self-supervised pre-training enables the learning of highly transferable visual representations without expert annotations, particularly when applied at scale (Chen et al., 2020; He et al., 2020; Siméoni et al., 2025). Models trained with discriminative self-supervised objectives have been reported to outperform conventional supervised pre-training on natural image datasets, especially in the presence of strong domain shifts (Boers et al., 2024; Perez-Garcia et al., 2025). In endoscopic image analysis, this paradigm offers a promising approach to mitigate the scarcity of expert-labeled data by leveraging large repositories of unlabeled clinical images. However, the effect of in-domain self-supervised pre-training on inflammatory disease detection in upper GI endoscopy has so far received limited attention. Therefore, this paper presents three key contributions to the field, as outlined below:

- We show that self-supervised pre-training on large-scale natural image datasets significantly improves performance on the esophagitis detection task, with additional performance gains achieved through self-supervised pre-training on an in-domain endoscopy dataset.

- We curate an esophagitis dataset from multiple public datasets encompassing images from three clinical centers, which closely reflects the conditions encountered during routine upper GI endoscopy.

- We release a model family of Vision Transformers pretrained with DINOv3 on a large-scale upper GI endoscopy image dataset, which can be used as visual backbones for further endoscopy-related downstream tasks.

## 2. Related Work

### 2.1. Esophagitis Detection

The detection of esophagitis using deep learning has become an active area of research in recent years, with the aim of supporting clinicians in the diagnostic process. Most existing approaches follow a common transfer learning paradigm, in which convolutional neural networks (CNNs) pretrained on large-scale natural image datasets are fine-tuned in a fully supervised manner for esophagitis-related classification tasks. While this strategy has shown promising results, it remains constrained by the substantial domain gap between natural and endoscopy images, as well as by the limited availability of annotated medical images.

Within this paradigm, a large body of work addresses esophagitis detection as a binary classification problem, distinguishing esophagitis from normal esophageal mucosa (Hou et al., 2021; Silveira et al., 2022; Yoshiok et al., 2023) or Barrett's esophagus (Mubarak et al.,

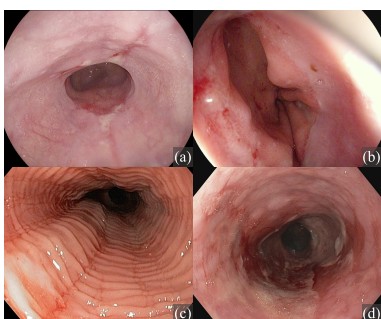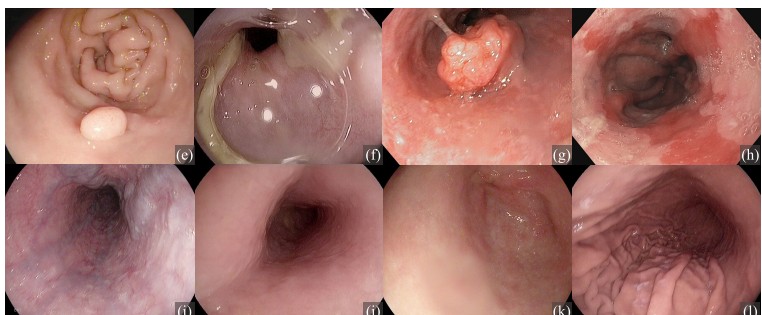

Figure 1: **Example images from the curated esophagitis dataset.** Images (a–d) show different manifestations of esophagitis and constitute the positive class for the downstream detection task. The negative class consists of other pathological findings, including polyps (e), ulcers (f), cancer (g), Barrett's esophagus (h), varices (i), as well as normal mucosa (j–l).

2022; Khullar et al., 2023). Closely related work focuses on eosinophilic esophagitis (EoE) detection (Guimarães et al., 2022; Okimoto et al., 2022; Römmele et al., 2022), a specific subtype of esophagitis.

Beyond binary formulations, several studies extend esophagitis recognition to multiclass classification settings that better reflect real-world clinical scenarios. These include joint classification of esophagitis with other gastrointestinal findings such as polyps, ulcers, cancer, or normal mucosa (Dheir and Abu-Naser, 2022; Tang et al., 2022; Singh and Sharma, 2024; Zubair Rahman et al., 2024; Krenzer et al., 2025).

Nevertheless, nearly all prior studies rely on supervised ImageNet pre-training and are evaluated under homogeneous or balanced dataset conditions, which limits conclusions about model robustness in heterogeneous clinical environments.

## 2.2. Self-Supervised Pre-training in Endoscopy

Self-supervised learning has emerged as a powerful paradigm for visual representation learning, enabling models to leverage large-scale unlabeled data through the use of surrogate supervision tasks. Building on its success in natural image domains (Chen et al., 2020; He et al., 2020; Siméoni et al., 2025), self-supervised pre-training has recently gained increasing attention in endoscopic image analysis.

Several studies have proposed foundation models pretrained on large collections of unlabeled endoscopic data. Wang et al. introduced EndoFM (Wang et al., 2023) and its extension EndoFM-LV (Wang et al., 2025), which exploit short and long video sequences from colonoscopy, gastroscopy, and laparoscopy. These models are pretrained using self-supervised objectives on video data and subsequently fine-tuned for downstream tasks such as polyp detection and segmentation in endoscopic videos.

In contrast, EndoViT (Batić et al., 2024) and the commercial EndoDINO (Dermyer et al., 2025) adopt an image-based approach and perform self-supervised pre-training on

individual endoscopic frames rather than video sequences. The resulting visual backbones have been evaluated on tasks including surgical video segmentation and anatomical landmark classification. While such approaches benefit from large quantities of training images, a notable limitation is the high degree of redundancy in the data: consecutive frames from endoscopic videos are often highly similar, and the number of distinct patients and anatomical variations represented in the dataset remains relatively limited.

Addressing this issue, recent work has explored self-supervised pre-training on endoscopic images extracted from clinical reporting systems that document routine examinations. Similar to the approach adopted in this study, Boers et al. (2024) and Devkota et al. (2025) utilized images that are automatically selected by gastroenterologists during standard clinical workflows. These images tend to exhibit greater visual diversity and emphasize clinically relevant findings, thereby providing a richer supervisory signal. Boers et al. (2024) demonstrated that this strategy yields strong performance gains for Barrett's neoplasia and esophageal cancer detection.

However, despite these advances, the application of image-based self-supervised pre-training to inflammatory disease detection remains largely unexplored. In particular, there is limited evidence on how targeted, in-domain self-supervised representations learned from upper GI endoscopy images affect the detection of esophagitis.

## 3. Method

We explore whether discriminative self-supervised pre-training on upper GI endoscopy images can improve downstream performance for esophagitis detection. To this end, we propose a two-stage training pipeline consisting of in-domain self-supervised pre-training followed by task-specific fine-tuning. An overview of the proposed training workflow is illustrated in Figure 2. We systematically compare our approach against two baselines: (i) supervised pre-training on natural images using ImageNet-1K, and (ii) large-scale self-supervised pre-training on the unlabeled LVD-1689M image dataset (Siméoni et al., 2025).

### 3.1. Pre-training on Endoscopy Images

#### 3.1.1. Data collection: UpperGI-400K

For in-domain pre-training, we retrospectively collect endoscopic images from the archive of the University Hospital of Würzburg. The images are acquired during upper GI endoscopy procedures. Data extraction is performed by the local IT department using predefined queries within the clinical endoscopy reporting system.

In total, 425,204 endoscopic images acquired between 2014 and 2025 from 21,535 unique patients have been collected. Throughout this paper, we refer to this private dataset as UpperGI-400K. The majority of images are acquired using endoscopy equipment produced by Olympus; however, images from Fujifilm and Pentax devices are also included. As the data originate from routine clinical reporting, the dataset contains images depicting both pathological findings and normal mucosa documented for clinical purposes. No image-level labels are available.

To prevent the self-supervised learning process from being biased by non-informative background regions, particularly the black borders commonly present in endoscopic images,

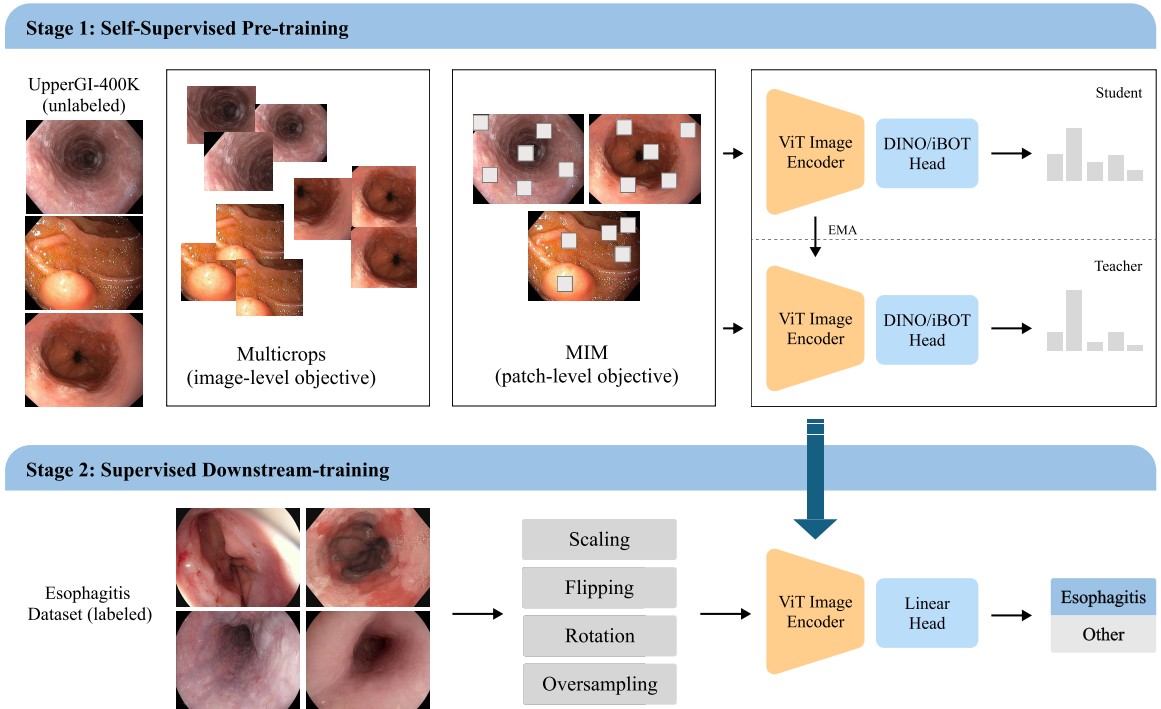

Figure 2: **Overview of the training pipeline employed in this study.** First, a ViT model is pre-trained using DINO self-supervised learning on a large, unlabeled in-domain dataset. The learned weights of the image encoder are then used for model initialization and subsequently fine-tuned in a supervised manner for the downstream task of esophagitis detection.

we apply an automatic cropping procedure. Specifically, we employ an intensity thresholding–based algorithm combined with morphological closure and contour extraction to isolate the largest contiguous region of interest. Irrelevant border regions are removed, ensuring that feature learning during pre-training focuses exclusively on clinically relevant visual content.

As the downstream esophagitis detection task is restricted to white-light imaging (WLI), we additionally filter out narrow-band imaging (NBI) images. For this purpose, RGB images are converted into the HSL color space, and each pixel is assigned a discrete color based on its hue value. Images are excluded if pixels corresponding to blue or green hues constitute more than 5% of the total image area. After applying this filtering step, the final pre-training dataset consists of 395,201 white-light endoscopic images.

### 3.1.2. DISCRIMINATIVE SELF-SUPERVISED PRE-TRAINING

We perform self-supervised pre-training using DINOv3 (Siméoni et al., 2025), which combines image-level discriminative learning with patch-level masked image modeling to learn rich visual representations from unlabeled images. The approach follows a teacher–student

architecture, where both networks share the same backbone and the teacher parameters are updated as an exponential moving average of the student parameters (He et al., 2020). Stage 1 of Figure 2 provides an overview of the DINO-based training procedure.

At the image level, global semantic representations are learned using the DINO loss, which aligns the output distributions of the student and teacher networks. Both networks are equipped with a DINO projection head, and training is performed on multiple strongly augmented views of the same image, including random global and local crops (Caron et al., 2021). To capture fine-grained spatial information, DINOv3 additionally incorporates a patch-level masked image modeling objective based on the iBOT approach (Zhou et al., 2022). A subset of image patches is randomly masked, and the student network predicts the teacher's representations for these masked patches using a dedicated iBOT head, promoting localized feature discrimination.

In contrast to Siméoni et al. (2025), we do not perform a subsequent high-resolution adaptation stage during pre-training, but instead keep the input resolution fixed at 224 throughout the training phase to ensure a fair comparison with the baseline models in the downstream detection task. We further omit the Gram anchoring step introduced in DINOv3, as this component primarily benefits segmentation tasks and our total training schedule is considerably shorter. Similar to Perez-Garcia et al. (2025), we adapt the dual-view data augmentation strategy to meet domain-specific requirements. Since lesion detection relies on both fine-grained texture and broader contextual information, we employ larger crop sizes and apply less aggressive blurring on the teacher branch.

We train Vision Transformer (ViT) models of sizes Small, Base, and Large on the UpperGI-400K dataset. All student networks are initialized with the corresponding DINOv3 pre-trained weights obtained from the LVD-1689M dataset, which consists of web images collected from Instagram posts (Siméoni et al., 2025). Training is conducted with a batch size of 128 for 125,000 iterations using two NVIDIA L40 GPUs with 48 GB memory each. Under this setup, training the ViT-L model requires approximately ∼2 days.

### 3.2. Esophagitis Detection

#### 3.2.1. Data Collection and Curation

To date, there is no single large-scale, publicly available dataset dedicated to esophagitis detection that could serve as a standardized benchmark. To address this gap, we curate a composite dataset that closely reflects the conditions encountered during routine upper GI endoscopy. For this purpose, we aggregate data from three publicly available GI endoscopy datasets with expert annotations: HyperKvasir (Borgli et al., 2020), ERS (Cychnerski et al., 2022), and GastroVision (Jha et al., 2023). By combining data from three different clinical centers, the resulting dataset captures a broader variability in acquisition devices, clinical practice, and patient populations, thereby increasing robustness of trained models against in-domain distribution shifts.

All source datasets contain images covering the entire GI tract. However, we exclusively consider images from the upper GI tract. This restriction is motivated by the anatomical and visual differences between upper and lower GI regions, for example between the esophagus and the colon. A model could easily infer that images depicting the colon cannot, by definition, show esophagitis. Including lower GI images would therefore substantially

increase the proportion of easy negative samples in the training and evaluation data that do not occur in the clinical setting of upper GI endoscopy and could lead to an overestimation of model performance.

The datasets exhibit considerable heterogeneity with respect to annotation granularity and labeling conventions. To enable a medically consistent and reliable merging of the datasets, we designed a flexible class-mapping framework that consolidates fine-grained labels into clinically meaningful higher-level categories. For the binary classification task addressed in this work, the positive class consists of images showing esophagitis, while the negative class includes normal mucosa as well as other pathological findings, such as varices, polyps, ulcers, cancer, and Barrett's esophagus. To ensure high label fidelity, we restrict our experiments to images with precise annotations in the ERS dataset. Representative samples from the curated esophagitis dataset are shown in Figure 1.

To further increase the number of esophagitis samples, we additionally collect images from publicly accessible endoscopy atlases and medical textbooks available online that are explicitly labeled as esophagitis. As these sources are commonly used for the education and training of gastroenterologists, we consider the provided labels to be reliable ground truth. We note that these textbook images are, on average, slightly lower in resolution than clinical endoscopy images. However, since almost all evaluated models operate on a fixed input resolution of $224 \times 224$, this difference is largely mitigated during preprocessing and is therefore unlikely to substantially affect downstream performance. This process adds 133 additional esophagitis images to the dataset. In total, the final curated dataset comprises 1,047 images of esophagitis and 5,822 other upper GI endoscopic images. Of the latter, 4,508 depict normal mucosa and 1,314 represent other pathological findings.

### 3.2.2. Training Procedure

For fine-tuning the pre-trained models on the esophagitis detection task, the dataset is split into 65% training, 15% validation, and 20% testing sets. We use a stratified group split strategy that preserves group integrity while maintaining class-level stratification. This ensures that samples from the same patient do not appear in both training and test sets, thereby preventing data leakage.

Due to the highly imbalanced esophagitis dataset, we apply oversampling during training to approximately balance the ratio of positive and negative samples within each mini-batch. In addition, we perform a set of data augmentations tailored to endoscopy images, including random horizontal and vertical flipping, cropping, scaling, and rotations by 90°, 180°, and 270°. We experimented with different intensities of color jittering and Gaussian blurring; however, these augmentations did not yield improvements on the validation set and were therefore omitted from the final training runs.

All models are fine-tuned using the AdamW optimizer with a batch size of 128 for a total of 100 epochs, including a 5-epoch warm-up phase. The learning rate is scheduled using a cosine decay strategy. To ensure a fair and rigorous comparison across models, we conduct a hyperparameter sweep on the validation set for each architecture, varying the learning rate over $\left[1 \times 10^{-5}, 5 \times 10^{-5}, 1 \times 10^{-4}, 5 \times 10^{-4}, 1 \times 10^{-3}, 5 \times 10^{-3}\right]$ and the weight decay over $\left[2 \times 10^{-5}, 2 \times 10^{-4}\right]$.

### 3.2.3. Evaluation

We systematically evaluate the performance of three differently sized ViT architectures under multiple pre-training strategies. Specifically, each ViT model is trained using (i) supervised pre-training on natural images, (ii) the default DINOv3 self-supervised weights trained on natural images, and (iii) self-supervised pre-training on the UpperGI-400K dataset proposed in this work.

In addition, we fine-tune the ViT-S model introduced by Boers et al. (2024), which was pre-trained using DINOv1 on the large-scale GastroNet-5M dataset, to benchmark against existing foundation models specifically developed for gastrointestinal endoscopy. Due to their widespread use in the esophagitis detection literature, we further include a set of CNN baselines pre-trained on ImageNet. These comprise VGG-16 (Dheir and Abu-Naser, 2022; Tang et al., 2022; Yen et al., 2022), ResNet50 (Hou et al., 2021; Mubarak et al., 2022; Okimoto et al., 2022; Römmele et al., 2022; Khullar et al., 2023), DenseNet-121 (Silveira et al., 2022; Guimarães et al., 2022), and EfficientNet-B4 (Ge et al., 2023; Singh and Sharma, 2024; Zubair Rahman et al., 2024; Krenzer et al., 2025).

All models are trained using five different random seeds to ensure robustness and reproducibility of results. Performance is evaluated on the held-out test set of the esophagitis dataset. An input resolution of $224 \times 224$ is used for all models, with the exception of EfficientNet-B4, which operates at an input resolution of $320 \times 320$.

Given the class imbalance typical of medical imaging datasets, we evaluate model performance primarily using the Area Under the Precision–Recall Curve (AUPRC), which better reflects performance on the clinically relevant positive class. For completeness and comparability with prior work, we additionally report the Area Under the Receiver Operating Characteristic Curve (AUROC) and the F1 score.

## 4. Results and Discussion

Table 1 presents a comprehensive quantitative comparison of model architectures and pre-training strategies for the task of esophagitis detection. Across all architectures initialized with supervised ImageNet pre-training, performance remains consistently moderate. CNNs achieve AUPRC values between 77.63% and 83.03%, with corresponding F1 scores ranging from 72.96% to 75.54%. Increasing model capacity using Vision Transformers yields only marginal improvements, with ViT-L/16 reaching an AUPRC of 84.31%. These results indicate that, despite architectural advances, supervised pre-training on natural images alone is insufficient to fully capture the visual characteristics required for robust esophagitis detection in heterogeneous upper GI endoscopy data.

Replacing supervised ImageNet pre-training with self-supervised alternatives already leads to noticeable performance gains, as this removes the reliance on labeled data and enables the use of substantially larger datasets. While DINOv1 pre-training on ImageNet-1K does not yield consistent improvements, scaling self-supervised pre-training to large-scale natural image data using DINOv3 substantially enhances downstream performance. ViT models pretrained with DINOv3 on LVD-1689M improve AUPRC by up to 5.44 percentage points compared to their supervised counterparts, with ViT-L/16 reaching an AUPRC of 87.23% and an F1 score of 80.83%. These findings highlight the benefit of discriminative self-supervised representations, even when pre-training on out-of-domain data.

Table 1: **Quantitative evaluation of model architectures and pre-training techniques on the esophagitis detection task.** Reported values are mean ± standard deviation over 5 seeds. Higher values indicate better performance.

| Model | Pre-training | | Esophagitis detection | | |
| --- | --- | --- | --- | --- | --- |
| | Data set | Method | AUPRC | AUROC | F1 |
| *Supervised on natural images* | | | | | |
| VGG-16 | ImageNet-1K | Supervised | $83.03 \pm 2.65$ | $95.06 \pm 1.19$ | $75.54 \pm 2.29$ |
| ResNet50 | ImageNet-1K | Supervised | $77.63 \pm 1.76$ | $93.17 \pm 2.10$ | $74.10 \pm 2.29$ |
| Densenet-121 | ImageNet-1K | Supervised | $80.89 \pm 5.11$ | $94.33 \pm 1.37$ | $72.96 \pm 4.42$ |
| EfficientNet-B4 | ImageNet-1K | Supervised | $80.14 \pm 1.65$ | $93.73 \pm 0.35$ | $74.85 \pm 1.74$ |
| ViT-S/16 | ImageNet-1K | Supervised | $83.23 \pm 1.26$ | $94.46 \pm 1.46$ | $75.57 \pm 2.40$ |
| ViT-B/16 | ImageNet-1K | Supervised | $82.40 \pm 4.15$ | $94.51 \pm 2.48$ | $75.64 \pm 2.92$ |
| ViT-L/16 | ImageNet-1K | Supervised | $84.31 \pm 2.06$ | $94.98 \pm 1.06$ | $76.74 \pm 1.34$ |
| *Self-Supervised on natural images* | | | | | |
| ViT-S/16 | ImageNet-1K | DINOv1 | $82.51 \pm 3.05$ | $95.88 \pm 0.73$ | $67.75 \pm 5.85$ |
| ViT-S/16 | LVD-1689M | DINOv3 | $88.67 \pm 0.93$ | $96.54 \pm 1.02$ | $79.05 \pm 1.59$ |
| ViT-B/16 | LVD-1689M | DINOv3 | $85.35 \pm 1.01$ | $96.15 \pm 0.63$ | $77.36 \pm 2.30$ |
| ViT-L/16 | LVD-1689M | DINOv3 | $87.23 \pm 2.48$ | $96.47 \pm 0.99$ | $80.83 \pm 1.64$ |
| *Self-Supervised on endoscopy images* | | | | | |
| ViT-S/16 | GastroNet-5M | DINOv1 | $89.29 \pm 0.80$ | $96.70 \pm 0.08$ | $79.25 \pm 1.01$ |
| ViT-S/16 | UpperGI-400K | DINOv2/3 | $86.16 \pm 1.53$ | $96.64 \pm 0.34$ | $77.47 \pm 1.40$ |
| ViT-B/16 | UpperGI-400K | DINOv2/3 | $88.88 \pm 0.92$ | $97.04 \pm 0.57$ | $80.15 \pm 0.93$ |
| ViT-L/16 | UpperGI-400K | DINOv2/3 | $\mathbf{89.82 \pm 0.68}$ | $\mathbf{97.17 \pm 0.34}$ | $\mathbf{81.44 \pm 0.74}$ |

Self-supervised pre-training further benefits from being performed on endoscopy-specific data. ViT models pre-trained with DINO on large in-domain datasets consistently outperform both supervised and self-supervised natural image baselines across all evaluation metrics. In particular, ViT-L/16 pretrained on the UpperGI-400K dataset achieves the best performance, with an AUPRC of 89.82%, an AUROC of 97.17%, and an F1 score of 81.44%. Compared to supervised ImageNet pre-training, this corresponds to an average improvement of 6.60 percentage points in AUPRC and 2.44 percentage points in AUROC. Notably, similar performance is achieved when pre-training on the ∼12 larger GastroNet-5M dataset, suggesting that both dataset scale and domain relevance contribute to effective representation learning for downstream endoscopy tasks. The relatively small performance gap compared to the smaller but anatomically focused UpperGI-400K dataset indicates that targeted in-domain curation and larger models can compensate for smaller dataset sizes in self-supervised pre-training.

In the context of prior work, these results suggest potential limitations of existing esophagitis detection models. Previous studies have reported accuracies exceeding 94% and AUROC values above 96% using supervised ImageNet-pretrained CNNs (Hou et al.,

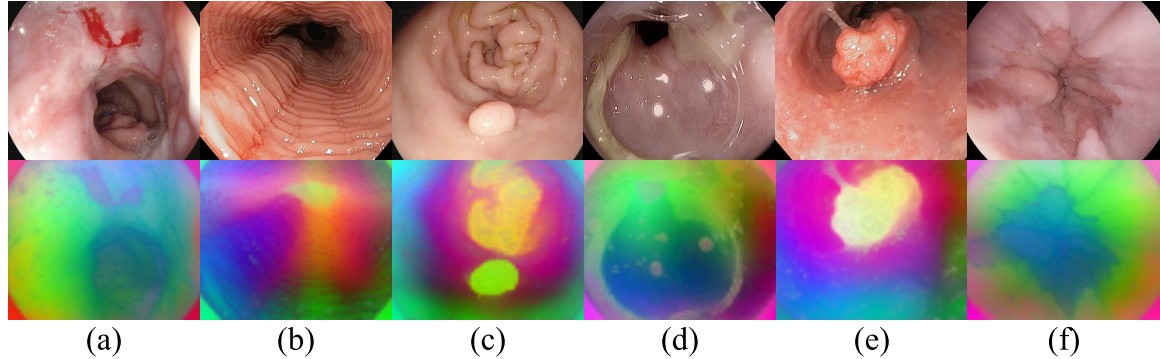

Figure 3: Feature embeddings extracted from ViT-L/16 after pre-training with DINO on the UpperGI-400K dataset, visualized by projecting the first three principal components into RGB space. Columns (a) and (b) correspond to esophagitis, while columns (c) polyp, (d) ulcer, (e) cancer, and (f) Barrett's esophagus.

2021; Silveira et al., 2022; Mubarak et al., 2022; Römmele et al., 2022). However, such performance is often reported in comparatively constrained experimental settings, including small datasets, simplified classification tasks, balanced class distributions, and limited variability among negative samples. In contrast, our evaluation is performed on a more challenging multi-center dataset with diverse confounding pathologies, leading to reduced performance for supervised baselines. The observed performance gap suggests that prior results may overestimate real-world performance. Consistent with recent findings on limited generalization of supervised transfer learning in complex endoscopic settings (Krenzer et al., 2025), our results demonstrate that in-domain self-supervised pre-training offers an effective strategy to address these limitations.

In Figure 3, we visualize the results of a principal component analysis (PCA) of patch-level feature embeddings extracted by our self-supervised ViT-L/16 model to gain insight into the learned representations of endoscopy images. Across all examples, low-informative regions such as black borders and dark luminal openings are consistently grouped into a distinct pink cluster, indicating effective separation of irrelevant image areas. Clinically relevant tissue patterns are well captured: red inflammatory regions are clearly separated from healthy mucosa (a), while characteristic furrows of EoE (b) are reflected by gradual intensity changes in the PCA maps. Protruding structures such as polyps (d) and tumors (e) are accurately delineated from surrounding tissue. In addition, salmon-colored mucosal extensions typical of Barrett's esophagus (f) are distinctly separated from the adjacent squamous epithelium of the esophagus. This indicates that the self-supervised model learns semantically meaningful representations aligned with clinically relevant endoscopic structures, despite the absence of pixel-level supervision.

To evaluate the data efficiency of self-supervised pre-training with a targeted in-domain dataset such as UpperGI-400K, we analyzed how performance changes as a function of the available labeled training data. To this end, we progressively reduced the training set size of the esophagitis dataset to 50%, 25%, 10%, and 5%, while keeping the validation and

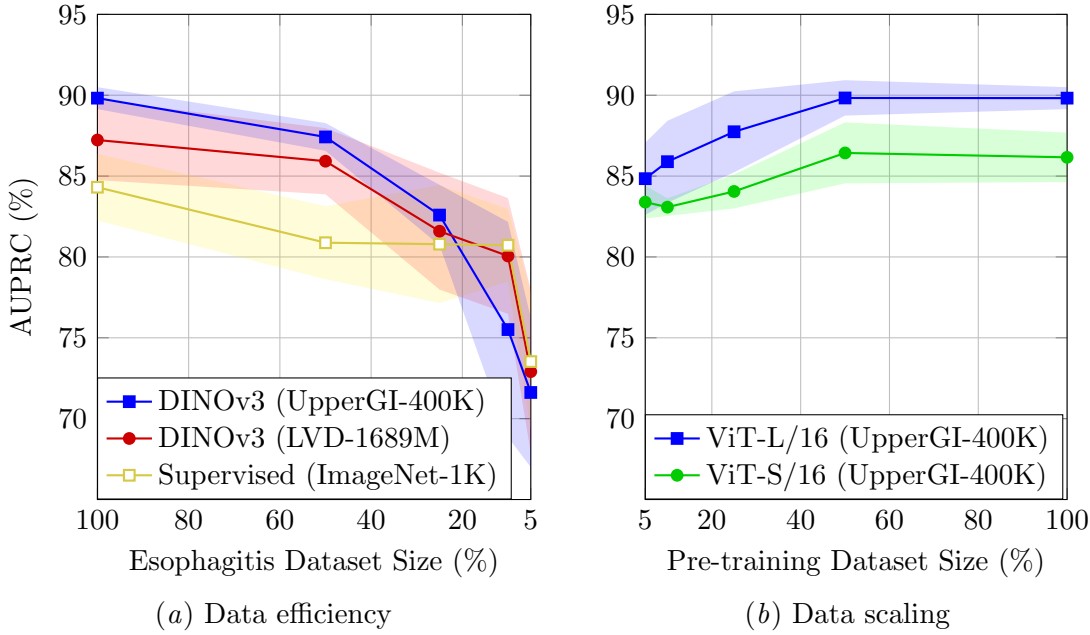

(a) Data efficiency    (b) Data scaling

Figure 4: **Data efficiency of models on the esophagitis detection task and data scaling during self-supervised pre-training.** Performance is evaluated using varying fractions of the available datasets. Each line reports the mean AUPRC over five random seeds, measured on the downstream esophagitis detection task. (a) Comparison of different pre-training strategies for a ViT-L/16 model with a stepwise reduction of the downstream training set size. (b) Data scaling effect of the in-domain pre-training dataset size (UpperGI-400K) on downstream performance.

test sets unchanged. At each data fraction, we measured the downstream performance of a ViT-L/16 model under three different pre-training strategies: supervised pre-training on natural images (ImageNet-1K), self-supervised pre-training on natural images (LVD-1689M), and self-supervised pre-training on endoscopy images (UpperGI-400K). The model pre-trained on UpperGI-400K consistently achieves the best performance from a dataset size of 25% and above. Notably, at 50% of the downstream training data, it reaches an AUPRC of 87.42%, already surpassing all other pre-training strategies trained with 100% of the dataset. For very small training set sizes (10% and below), we observe mixed results, with the supervised ImageNet-pretrained model being slightly superior on average. However, no strong conclusions should be drawn in this regime, as the 10% setting comprises only 446 training samples (∼68 positive and ∼378 negative), making the results susceptible to random variation and rapid overfitting. Results are visualized in Figure 4(a).

Furthermore, we studied the data scaling behavior of the proposed pre-training strategy. To this end, we systematically increased the size of the UpperGI-400K dataset and subsequently evaluated the downstream performance for esophagitis detection using ViT-S/16

and ViT-L/16 architectures. The performance trends of both models are highly similar. Notably, with only 5% of the pre-training data (∼20,000 images), the ViT-L/16 model achieves an AUPRC of 84.84%, outperforming all baseline models that are pre-trained on ImageNet using supervised learning. Increasing the number of training samples leads to a gradual performance improvement up to 50% of the dataset. Beyond this point, the performance saturates, and no further gains in esophagitis detection are observed with additional data. This suggests that the model captures the relevant variability of endoscopic images from this center and learns sufficiently expressive representations already with 50% of the available data. Collecting and incorporating data from multiple centers with heterogeneous patient populations and varying acquisition protocols may potentially yield further performance improvements when scaling beyond 200,000 images. Results are displayed in Figure 4(b).

While the proposed in-domain self-supervised pre-training leverages a large corpus of unlabeled upper GI endoscopy images, the pre-training data originate from a single center. Although images are acquired using devices from multiple manufacturers and span more than a decade, the learned representations may still reflect institution-specific imaging protocols or patient populations. Evaluating the pre-trained models on additional external datasets and tasks could further assess generalization. In addition, the downstream esophagitis dataset, despite being curated from multiple public sources and including a diverse set of confounding pathologies, remains limited in absolute size compared to large-scale natural image benchmarks. While this reflects the current availability of labeled esophagitis data, larger, prospectively collected datasets would enable more comprehensive evaluation.

## 5. Conclusion

In this work, we studied discriminative self-supervised pre-training for esophagitis detection in upper GI endoscopy. We systematically compared supervised ImageNet initialization with large-scale self-supervised pre-training on natural images and in-domain self-supervised pre-training on upper GI endoscopy data. Our results show that self-supervised pre-training consistently improves downstream performance, with the strongest gains achieved when representations are learned directly from endoscopy images.

Additional analyses indicate that in-domain self-supervised pre-training improves data efficiency, enabling strong downstream performance even with reduced amounts of labeled training data. Furthermore, data scaling experiments suggest that most performance gains for esophagitis detection can be achieved with a moderate number of in-domain pre-training images, while additional data from a single center leads to diminishing returns.

These findings highlight the importance of in-domain data collection and self-supervised learning as key enablers of enhanced accuracy in endoscopic image analysis, paving the way for more generalizable AI systems in clinical practice.

## Acknowledgments

This research project is funded by the Bavarian Research Institute for Digital Transformation (bidt), an institute of the Bavarian Academy of Sciences and Humanities. Additionally, this work was supported by the bayerischen tschechischen Hochschulagentur (BTHA) under grant BTHA-JC-2024-52. The author is responsible for the content of this publication.

## Ethical Considerations

The study including retrospective collection of examination images was approved by the responsible institutional review board (Ethical committee Würzburg, 12 August 2025, 2025-325-dvhd). All images used in this study were fully anonymized prior to use, with any personally identifiable information irreversibly removed in accordance with ethical and data protection standards.

## Code availability

The training and inference scripts, along with the trained models, have been made publicly available at https://github.com/tofriede/SSL-Esophagitis-Detection .

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
