# OpenReview forum: "Discriminative Self-Supervised Pre-Training for Esophagitis Detection in Upper GI Endoscopy Images"
_MIDL.io/2026/Conference — MIDL 2026 Poster_

### Official Review · Reviewer_64Zs · 2026-01-10

**Confidence:** 4
**Preliminary Rating:** 4
**Final Rating:** 4

**Summary:**

This manuscript investigates discriminative self-supervised pre-training for esophagitis detection in upper GI endoscopy images. The authors pre-trained Vision Transformers using DINOv3 on a large-scale, unlabeled in-domain dataset (UpperGI-400K) and evaluate downstream performance on a curated, multi-center esophagitis dataset. Through systematic comparisons against supervised ImageNet pre-training, large-scale self-supervised natural image pre-training, and existing GI foundation models, the study demonstrates that in-domain self-supervised pre-training yields consistent and meaningful performance gains. The work is well-motivated, carefully executed, and addresses an important practical limitation in clinical endoscopy AI: scarcity of expert annotations and poor generalization of supervised transfer learning.

**Strengths:**

The paper clearly articulates why esophagitis detection is clinically important and why annotation scarcity and domain shift limit current supervised approaches. The focus on inflammatory diseasefills a notable gap in the endoscopy AI literature.

The authors perform a well-structured comparison across several baselines. This strengthens the validity of the conclusions.

The curated esophagitis dataset includes diverse negative classesand is constructed from multiple centers. This setup is more clinically realistic than many prior works and likely explains the more conservative—but credible—performance numbers.

The PCA visualization of patch-level embeddings provides convincing qualitative evidence that the self-supervised model learns semantically meaningful and clinically relevant representations.

The paper clearly describes training details, hyperparameter sweeps, data splits, and provides a public code repository.

**Weaknesses:**

Although UpperGI-400K is large and diverse in time span and device vendors, it originates from a single institution. This raises concerns about potential institutional bias in learned representations. While acknowledged by the authors, the limitation remains important.

Despite careful curation, the downstream esophagitis dataset remains relatively small (∼1k positives). This may limit the statistical power of comparisons and the generalizability of conclusions.

Esophagitis is treated as a single positive class. Clinically, severity grading (e.g., LA classification) or subtype differentiation (e.g., reflux vs. eosinophilic) could be highly relevant. The current formulation simplifies the problem but also limits clinical interpretability.

While the dataset is multi-center, all evaluation is performed within a single curated benchmark. A true external validation (e.g., leave-one-dataset-out testing) would further strengthen claims about robustness.

The main novelty lies in the application and empirical validation of existing self-supervised methods (DINOv3) in a new clinical context, rather than in proposing new algorithms.

**Detailed Comments:**

While a stratified group split is used, a brief clarification on how patient IDs were handled across the merged public datasets would improve clarity and reassure readers about leakage prevention.

Even a small exploratory analysis (e.g., reflux vs. EoE, or mild vs. severe esophagitis) could substantially increase the clinical impact.

Consider adding a short ablation or discussion on how many pre-training images are needed to achieve most of the gains .

Although the focus is image-based, a brief discussion contrasting image-based vs. video-based SSL approaches (e.g., EndoFM) in terms of practicality and annotation cost would be valuable.

**Justification Of Final Rating:**

This paper presents a well-executed and carefully validated empirical study demonstrating that in-domain discriminative self-supervised pre-training substantially improves esophagitis detection performance in upper GI endoscopy images. The work is methodologically sound, clinically relevant and with a strong emphasis on realistic evaluation settings. Empirically, the paper provides consistent and non-trivial performance gains, supported by both quantitative results and qualitative feature analyses. The findings align well with emerging evidence that domain relevance can partially compensate for dataset scale in medical imaging foundation models.

**Justification Of The Preliminary Rating:**

This paper presents a well-executed and carefully validated empirical study demonstrating that in-domain discriminative self-supervised pre-training improves esophagitis detection performance in upper GI endoscopy images. The work is methodologically sound, clinically relevant and with an emphasis on realistic evaluation settings. Empirically, the paper provides consistent performance gains, supported by both quantitative results and qualitative feature analyses. The findings align well with emerging evidence that domain relevance can partially compensate for dataset scale in medical imaging foundation models.

**Questions To Address In The Rebuttal:**

How well do representations pre-trained on single-center UpperGI-400K data generalize to truly external institutions with different acquisition protocols? Can the authors clarify whether any leave-one-dataset-out or external-only evaluation is feasible, or provide justification for why the current setup sufficiently demonstrates robustness?

How were patient identities handled across merged public datasets, and can the authors confirm that no patient-level leakage occurs between train/val/test splits after dataset aggregation? Are there systematic differences between images sourced from public atlases/textbooks and clinical datasets (e.g., image quality, disease severity), and could this introduce bias in the positive class?

---

> ### Author Response · Authors · 2026-01-24
>
> We thank the reviewer for the careful assessment of our work and for the constructive suggestions.
>
> 1. Regarding generalization beyond the single-center UpperGI-400K pretraining data, we acknowledge that pretraining on data from multiple institutions would be the strongest form of robustness validation. However, we would like to emphasize that all downstream evaluations are conducted on a curated esophagitis dataset composed exclusively of external data from multiple public sources, including HyperKvasir, ERS, GastroVision, and publicly accessible atlases and textbooks. None of these images were used during SSL pretraining. As such, the evaluation already reflects transfer to institutions with different acquisition protocols, patient populations, and annotation standards. While a formal leave-one-dataset-out evaluation would be an interesting extension, the limited size of individual datasets and the heterogeneous labeling schemes make such an analysis challenging at this stage. We therefore believe that the current setup provides a meaningful and practical assessment of cross-institution generalization, while recognizing that broader external validation remains an important direction for future work.
>
> 2. Concerning patient identity handling and leakage prevention, we confirm that no patient-level leakage occurs between training, validation, and test splits. Each clinical dataset was collected from different hospitals located in different countries. In theory, it would be possible that the same patient underwent identical or very similar endoscopic examinations in multiple countries. However, such a scenario is extremely unlikely in practice and can reasonably be neglected in this context. We therefore consider the risk of cross-dataset patient overlap to be negligible, and the applied splitting strategy to be sufficient to prevent patient-level leakage.
>
> 3. We acknowledge that the downstream esophagitis dataset, with approximately 1,000 positive samples, is relatively small. This reflects the current scarcity of publicly available, well-annotated endoscopy data and may limit statistical power for fine-grained comparisons between top-performing models. Nevertheless, we observe consistent trends across paired runs with identical random seeds, which increases confidence in the reported performance differences. We agree that larger labeled datasets would further strengthen conclusions and view this as an open challenge for the community.
>
> 4. With respect to the binary formulation of esophagitis detection, we agree that severity grading, such as LA classification, or subtype differentiation, such as reflux versus eosinophilic esophagitis, would increase clinical interpretability. Our decision to focus on binary detection was motivated by label availability and consistency across public datasets. We agree that even a coarse severity or subtype analysis would be valuable and plan to investigate such extensions as more suitable annotations become available.
>
> 5. In response to the reviewer’s comment on scaling behavior, we have added an explicit data scaling analysis showing how downstream performance evolves as a function of the number of SSL pretraining images. These results demonstrate that most performance gains are achieved up to approximately 200K pretraining images, with diminishing returns beyond that point when adding more data from the same center. This analysis is now included in Section 4.
>
> 6. Regarding potential systematic differences between images sourced from public atlases or textbooks and those from clinical datasets, we note that textbook images are on average slightly lower in resolution. However, since almost all evaluated models use a fixed input resolution of 224 x 224, this difference is largely mitigated during preprocessing and is unlikely to substantially affect downstream performance. With respect to disease severity, medical textbooks aim to train clinicians on both clear and subtle manifestations of disease, including early-stage cases. We therefore expect the severity distribution in textbook images to largely overlap with that of the clinical datasets and do not believe that their inclusion introduces a meaningful bias in the positive class. We have now addressed the difference in section 3.2.1.

---

### Official Review · Reviewer_PwdB · 2026-01-15

**Confidence:** 4
**Preliminary Rating:** 5
**Final Rating:** 5

**Summary:**

This is a well-written application paper on esophagitis detection in endoscopic images. The authors have extracted a huge dataset of relevant images (not videos) from their clinical systems and used them for a DINOv3-based pre-training of ViT-based models that perform binary esophagitis detection (ignoring other pathologies). For testing, they curated a (public?) test set from public sources, spanning at least three clinical centers as well as a few hundred images from textbooks. The proposed approach (the resulting models of which are also published) is shown to work substantially better than reference models which are only pre-trained on ImageNet or self-supervised on the LVD-1689M dataset. The results also show that there is a relatively small performance gap between the proposed method and a comparison model that was pre-trained on the GastroNet-5M dataset which is ca. 12 times larger but less anatomically focused on the upper GI tract, hinting at the relative contributions of dataset size and domain match.

**Strengths:**

At first sight, the contributions do not read novel – it is well-known that self-supervised pretraining improves performance. Reading the full paper however, the authors have provided a good overview of the state of the art, the experiments are comprehensive, and a number of well-motivated design choices are made that go beyond simply throwing an existing method on ones own data. The evaluation is really solid, both in the sizes of datasets, the availability of the test set, the data curation applied, and the measures chosen. The code and the models are also released.
The collection of about 400k images from about 20k patients used for pre-training is a sizeable batch.  The resulting models should also be interesting to the community.  Curating its data with a simple, reproducible narrow-band imaging filtering (about 7% of the raw data collection) is one of the aspects of good technical execution.
I also like the conclusions made from the observations, as well as the PCA visualizations of the feature embeddings, which are not important for this paper, but still a nice addition.

**Weaknesses:**

The methodological contributions are small, but I would still rate them solid, because they consist of well-motivated design choices.
I am not sure if the test set is published, or if the provided code allows to reproduce the data curation applied to the three public datasets in order to reproduce the test set (also considering the textbook images).

**Detailed Comments:**

I suggest to rephrase the results in the abstract, "performance gain of 6.60% in AUCPRC" is ambiguous w.r.t. percent vs. percent points (the latter is meant, as one can see from the unambiguous main manuscript), and the relative change is hard to assess without knowing the absolute values before / after. I suggest to state this properly and remove the AUROC improvement from the abstract if it otherwise becomes too long.
I also thought a lot about the statement that "replacing supervised ImageNet pre-training with self-supervised alternatives already
leads to noticeable performance gains"; I think that is because the gains do not come from self-supervision vs. supervision, but from a) the much larger datasets being used for training, and from b) the training scheme using advanced augmentation and losses, as well as the student-teacher approach. All of that could theoretically be combined with supervised training as well (if one had enough labels). This is just implied, and somehow the sentence read wrong to me as it stands now. Probably a small tweak could alleviate that feeling.
I found it very interesting that the authors "additionally collect images from publicly accessible endoscopy atlases and medical textbooks available online". On the one hand, this is an awesome idea, because this is what human doctors also learn from, so the examples should be well-curated (and this argument is also made within the article), on the other hand I wonder about copyrights and legal discussions about the use for training AIs. I also wonder if the test data is published, or if that part of the data collection is reproducible for the community.
I find the paper quite well-written and found no language problems. I stumbled over "teacher-student architecture", because I have always been hearing it as "student-teacher", but apparently both are used in general.
You state "to assess practical clinical usage, we report the F1 score computed at a fixed decision threshold of 0.5", but I am not sure I can follow you a) why the F1 score is the most relevant one for clinical usage, and b) why 0.5 is the best operating point. I would have thought that other operating points may be more suitable, given that medical doctors will probably review the AI results anyhow.

**Justification Of Final Rating:**

I already rated this manuscript very positively, and it has only improved since then. The points raised by the other reviewers were also addressed sufficiently from my point of view. I will maintain my original rating.

**Justification Of The Preliminary Rating:**

This is an excellent example of an application paper. The evaluation is strong and convinced me that the proposed method contributes to the state of the art. The code and model are published as well, increasing the value of this work for the community and the MIDL audience.

**Questions To Address In The Rebuttal:**

I would really like to understand whether the test set that is curated from three public datasets as well as publicly available text book images is reproducible for others or not?
Did you do any kind of statistical significance tests? In particular the "small performance gap" between the proposed method and the GastroNet-5M model would be interesting to test for significance, I think.

---

> ### Author Response · Authors · 2026-01-24
>
> We thank the reviewer for the careful reading of our manuscript and for the constructive and thoughtful feedback.
>
> 1. Regarding reproducibility of the curated esophagitis test set, the dataset can be largely reconstructed using the files `train_val.csv` and `test.csv` provided in our public repository. Since HyperKvasir, ERS, and GastroVision are all publicly accessible datasets, the data curation process is reproducible for the community. The relevant files and instructions are available at
> [https://github.com/tofriede/SSL-Esophagitis-Detection/tree/main/esodetector/data](https://github.com/tofriede/SSL-Esophagitis-Detection/tree/main/esodetector/data).
> The only exception are the textbook images, which cannot be redistributed due to licensing restrictions. However, these constitute only 133 samples, and we do not expect their exclusion to result in substantial deviations in evaluation outcomes.
>
> 2. With respect to the abstract, we agree that the phrasing “performance gain of 6.60% in AUCPRC” was ambiguous. In the revised version, we now explicitly state that this refers to an absolute improvement measured in percentage points and additionally report the AUCPRC value of the best performing model for better interpretability. To improve clarity and conciseness, we have also removed the AUROC improvement from the abstract, as suggested.
>
> 3. We also appreciate the reviewer’s nuanced interpretation of the observed self-supervised learning gains. We agree that these improvements should not be attributed solely to the absence of supervision, but rather to the ability to leverage substantially larger training datasets in combination with advanced training schemes. In the revised manuscript, we have rephrased the corresponding statement to better reflect this interpretation and to avoid implying that supervision itself is the limiting factor.
>
> 4. Concerning the use of the F1 score at a fixed decision threshold of 0.5, our original intention was to report a commonly used and intuitive operating point that balances precision and recall and allows straightforward comparison across models. We agree, however, that clinical deployment would likely require different operating points depending on the intended use case and the acceptable trade-off between false positives and false negatives. To avoid suggesting direct clinical applicability of this specific threshold and metric, we have removed the wording related to “practical clinical usage” from the revised manuscript.
>
> 5. Finally, regarding statistical significance, we did not perform explicit hypothesis tests. However, for each model we conducted paired experimental runs using the same five training seeds, ensuring consistent and fair comparisons across methods. We note that GastroNet-5M also belongs to the category of models pretrained with self-supervised learning on endoscopy images, which further supports our central hypothesis that in-domain self-supervised pretraining provides a positive effect for esophagitis detection. While the performance gap between our method and GastroNet-5M is relatively small, both models consistently outperform alternatives pretrained outside the domain.

---

> > ### Comment · Reviewer_PwdB · 2026-01-28
> > **Good rebuttal, revision not yet online?**
> >
> > * My question about the reproducibility of the external test set has been positively answered, with (expected) exception for the textbook samples. On the one hand, I agree with the authors that the small number of these samples is not expected to make a big difference, on the other hand, they have of course argued in the manuscript that the quality of these samples is much higher (in terms of diversity and relevance) than of the rest. But 95% reproducibility is still very good.
> >   * Maybe the test sets should be looked at separately – that would allow 100% reproducibility for comparative evaluations of other works on the one set, and give additional insights on the difference between the test sets for readers of this manuscript?
> > * The point raised by the other reviewers – which I agree is a good one – that it is interesting to investigate the data efficiency by using smaller subsets of the data seems to have been addressed with additional experiments in the revision. This sounds great!
> >   * ~~Maybe I am blind, but did you already update the PDF? To me, section 4 and superficially the rest of the paper look exactly the same.~~ I was blind – I have found the revision attached to a comment further above.
> > * Statistical hypothesis testing was not performed nor added during the revision. Given that the evaluation is otherwise very convincing to me, I'll accept this.
> >
> > I have also read the comments to the other reviewers (and their reviews) and still think the authors did a great job.

---

> > > ### Comment · Reviewer_PwdB · 2026-01-28
> > >
> > > Now that I read the new additions:
> > > * "All models are trained using five different random seeds" -> "trained multiple times using the same five random seeds"?
> > > * In Fig. 4, I would move the captions for (a) and (b) below the resp. subfigure.
> > >
> > > Again, I like the addition.

---

> > > > ### Author Response · Authors · 2026-01-28
> > > >
> > > > We thank the reviewer for the positive assessment of the revisions and for the constructive and detailed feedback.
> > > >
> > > > Regarding the textbook images in the external test set: public redistribution of these images is unfortunately not possible due to copyright restrictions. However, we fully agree with the reviewer on the importance of reproducibility. Therefore, we would like to explicitly clarify that, upon reasonable request, we are happy to share the missing textbook images required to reconstruct the external test set at 100% completeness via direct email communication.
> > > >
> > > > Concerning the comment on random seeds, we acknowledge that our wording in the official comment (5.) was ambiguous. The intended meaning of “trained multiple times using the same five random seeds” is that all evaluated models were trained using the identical set of five different random seeds, ensuring a fair and directly comparable evaluation across methods. Concretely, the following seeds were used consistently for every model: 0, 42, 84, 126, and 168.
> > > >
> > > > With respect to Figure 4, we appreciate the suggestion to move the captions for subfigures (a) and (b) below the respective panels. We tested this adjustment with the current LaTeX template and found that multi-line subfigure captions lead to suboptimal formatting and reduced readability. However, we fully agree with the reviewer’s intention to improve clarity. In a potential camera-ready version, we will therefore explicitly label the subfigures as "(a) Data efficiency" and "(b) Data scaling", which makes the distinction immediately clear while preserving a clean layout.

---

> > > > > ### Comment · Reviewer_PwdB · 2026-02-03
> > > > > **Random seeds**
> > > > >
> > > > > FWIW, I don't think you need to specify the exact random seeds (here or in a revised manuscript), because making use of them requires *exactly* the same setup as yours, and otherwise, assuming a proper RNG, any random seed is as good as any other.

---

### Official Review · Reviewer_2sVw · 2026-01-16

**Confidence:** 4
**Preliminary Rating:** 4
**Final Rating:** 4

**Summary:**

The authors in this work have proposed to improve automated detection of esophagitis in upper GI endoscopy images by reducing reliance on scarce expert annotations. GI annotations are scarce and it thus becomes difficult to train large models on this data. Specifically, the authors aim to demonstrate that discriminative self-supervised pre-training on large-scale, "in-domain endoscopy images" leads to more robust and generalizable representations than conventional supervised pre-training on natural image datasets such as ImageNet or others.

To achieve and validate this, the authors propose a two-stage pipeline in which ViTs (S/B/L) are first pretrained using DINOv3 on a large, unlabeled, in-domain dataset of upper GI endoscopy images. To this end, they collected a dataset of ~400K upperGI images (UpperGI-400K), and subsequently fine-tuned the DINOV3 on a curated esophagitis detection dataset aggregated from three public sources.

The authors have systematically compared three approaches - 1) Direct fine-tuning of Imagenet based backbones ; 2) DINOV3 based SSL backbones; and 3) their proposed SSL DINOV3 on in-domain GI images and then fine-tuning of that.

**Strengths:**

The work addresses an inherent problem of not having labelled data for training large models and investigates whether and which kind of SSL, specifically the "in-domain" SSL can help to fine-tune models for downstream tasks more effectively. Below are the major strengths:

1. Strong motivation and clinical relevance - Esophagitis is a common and clinically important condition, and early detection can prevent progression to severe disease. The authors have very clearly motivated the clinical relevance of GI images used for esophagitis detection. The paper clearly articulates the limitations of current fully supervised approaches under annotation scarcity and domain shift, making a compelling case for SSL in this setting.

2. Clear experimental design - The two-stage pipeline (SSL pretraining -> SFT) is clear and reasonable to follow for this kind of setting. The authors conduct controlled comparisons across: CNNs vs. ViTs; Supervised vs. SSL pretraining;  and Natural-image vs. in-domain SSL. This systematic structure strengthens the validity of the conclusions.

3. Large-scale in-domain pretraining dataset - One of the major strengths of this work is the availablity of their private UpperGI-400K dataset (≈395k unlabeled images from >21k patients). Leveraging routine clinical reporting images rather than redundant video frames increases data diversity.

4. PCA overlay visualization - Apart from the quantitative performance metrics (which cover a range from AUROC, F1 to AUPRC (more important one), the authors have also demosntrated the qualitative value of the embeddings by overlaying the top 3 components of the embeddings extracted from their approach to the different conditions in the GI images. This gives better justification on why the embeddings may be better.

5. Reproducibility and openness - For this kind of large scale dataset pretraining, it is really important that the authors release their models and I really liked that the authors have already released their pretrained models and training code increases which is a great to resource to the community working in this domain.

**Weaknesses:**

Below are the weakness of this work.

1. Single-center pretraining data - Although UpperGI-400K is large, spans multiple devices and years and it is the most important asset for this paper and in-domain SSL to take place, it originates from a single clinical center. This raises concerns about institution-specific biases in imaging protocols or patient populations as well. The authors acknowledge this limitation but does not empirically test cross-institution generalization, which may be out of scope of this paper, but something that the authors should consider for expanding this work.

2. Data scaling experiments - I think the motivation of in-domain SSL is great and large scale data is needed for it. However, how "large" is the large scale? I would really like to know whether authors systematically tested the SSL approach on different scales of the data, for example, training only on 100K, then 200K, .... or in terms of 50K and seeing the trend in the performance. Currently, the number of 400K is arbitrary selected.

3. Downstream dataset size is small - Despite being curated from multiple public sources, the esophagitis detection dataset contains only ~1,000 positive samples. While this reflects real annotation constraints, it limits the statistical power of downstream evaluation and may still underrepresent rare or subtle disease presentations.

4. Binary class distribution - I understand that the task is primarily to detect esophagitis, and thus the authors have constructed the task a binary classification task, however, as GI may also have other classifications such as polyp, ulcers, etc. it would also have been test whether this can be a multi-class classification instead of binary class. This makes the task more challenging and it would be more justifiable to understand the performance of their approach. Did the authors give a try to this downstream task setting?

5. Missing recent GI foundation models - While the authors compare against GastroNet-5M and DINO-based models, it does not include some very recent or video-based GI foundation models (e.g., EndoFM-LV variants [https://github.com/med-air/EndoFM-LV]) in a deeper downstream comparison. The positioning relative to the rapidly evolving FM landscape could be clearer.

6. Focus on classification only as the downstream task - The work evaluates esophagitis detection as image-level classification. Tasks such as localization, segmentation, and severity grading, which are highly relevant in endoscopy are not explored and evaluated. To fairly demonstrate the usability of the SSL approaches, multiple downstream tasks are usually preferred. Have the authors tried any of the other downstream tasks in the GI domain?

**Detailed Comments:**

Below are some suggestions for improvement:

1. Evaluate cross-dataset or cross-center generalization - Testing the pretrained models on an external dataset not used during SSL pretraining would significantly strengthen claims of robustness and clinical applicability. It is still missing and the reliance on only one big dataset for pretraining needs to be checked and thoroughly evaluated. Something that the authors should concentrate in the future.

2. Extend to more fine-grained clinical tasks - Exploring multi-class esophagitis subtypes, severity grading, or lesion localization would better demonstrate the clinical value of the learned representations. It would also be more challenging for the approach to demonstrate this and it's a better test of the generalazability of the features.

3. Broader comparison with GI foundation models - Including or discussing additional SOTA GI foundation models, especially video-based ones, would clarify the relative strengths and limitations of image-based SSL. I think the proposed appraoch is already quite systematic, however, this comparison would further strengthen the work.

4. Analyze data efficiency more explicitly - A learning-curve analysis showing performance as a function of labeled data size would further highlight the benefits of in-domain SSL under annotation scarcity.

5. Discuss deployment considerations - Additional discussion on computational cost, inference speed, and integration into real-time endoscopy workflows would enhance translational relevance.

**Justification Of Final Rating:**

The authors have addressed my comments in detail by running additional experiments on data efficiency and scaling which were missing in the original version. The results are now more in-depth. I really hope that this dataset could be released by the authors for the community and the authors could evaluate on multiple and more challenging downstream tasks and diverse datasets in the future to further improve this work. Thus, I stand by my original rating of acceptance.

**Justification Of The Preliminary Rating:**

This work presents a solid, well-executed study that convincingly demonstrates the value of in-domain discriminative SSL pretraining for esophagitis detection in upper GI endoscopy. While the methodological novelty is incremental (building on DINOv3 rather than introducing a new SSL objective), the clinical grounding, dataset scale, and thorough empirical evaluation make this a meaningful contribution. The authors are encouraged to improve their work based on the comments.

**Questions To Address In The Rebuttal:**

Please check the weakness and detailed comments to be addressed.

---

> ### Author Response · Authors · 2026-01-24
>
> We thank the reviewer for the constructive and insightful comments.
>
> 1. While the SSL pretraining dataset UpperGI-400K originates from a single clinical center, it is important to emphasize that the downstream esophagitis detection dataset consists exclusively of endoscopic images collected from multiple external institutions across different countries. None of these images were used during SSL pretraining. Consequently, the pretrained representations are evaluated on a fully external dataset that was not used during SSL, which in our view already constitutes a cross-center evaluation and provides evidence of generalization beyond the pretraining institution. We agree, however, that pretraining on data from multiple centers would further strengthen robustness claims and represents an important direction for future work.
>
> 2. In response to the reviewer’s suggestion regarding data scaling, we have conducted additional SSL pretraining experiments using progressively larger subsets of the UpperGI-400K dataset and added a corresponding analysis in Section 4. For both ViT-S/16 and ViT-L/16, downstream esophagitis detection performance improves consistently up to approximately 200K pretraining images. Beyond this point, adding more images from the same center does not lead to statistically significant performance gains, indicating a saturation effect when scaling data from a single institution.
>
> 3. We acknowledge that the esophagitis dataset, with approximately 1,000 positive samples, is relatively small compared to datasets in other computer vision domains. This limitation reflects the current scarcity of publicly available and well-annotated endoscopy data. We believe that addressing this challenge requires continued community efforts to curate and release larger labeled gastrointestinal datasets. We added a corresponding statement at the end of the discussion section (4).
>
> 4. Our study intentionally focuses on esophagitis detection as a clinically relevant and well-defined use case. We agree that broader downstream evaluations, such as multi-class classification across the upper gastrointestinal tract or fine-grained lesion characterization, would further demonstrate the generalizability of the learned representations. We therefore view such evaluations as a natural extension of this work and plan to investigate them in future research.
>
> 5. Regarding recent gastrointestinal foundation models, EndoFM and EndoFM-LV are specifically designed for video-based analysis. While it is technically possible to adapt these models to image-based classification by treating a single image as a one-frame video, the learned temporal attention mechanisms would not be meaningfully utilized in this setting. In addition, both models are pretrained on HyperKvasir videos, which overlap with the public datasets used in our downstream evaluation. This introduces the risk that samples from the test set may have been seen during pretraining. A fair and clean comparison would therefore require a different external test dataset. Rather than viewing these approaches as competing, we consider them complementary. Our results support the broader conclusion that large-scale in-domain self-supervised learning is highly effective for endoscopy data, not only for videos but also for still images. An interesting direction for future work is to systematically investigate how well video-based representations transfer to image-based tasks and vice versa.
>
> 6. As suggested by the reviewer, we have additionally included data efficiency experiments by systematically reducing the size of the labeled esophagitis training set. The resulting learning curve analysis is presented in Section 4.
>
> 7. Finally, since our work does not introduce a novel model architecture but relies on standard Vision Transformer backbones, we did not originally include a dedicated deployment section. For reference, our strongest model, ViT-L/16 with approximately 300 million parameters, achieves an inference throughput of approximately 250 images per second on an NVIDIA L40 GPU using a batch size of 64. As expected, inference speed depends on hardware, batch size, numerical precision, and framework-level optimizations.

---

> > ### Comment · Reviewer_2sVw · 2026-01-28
> >
> > I thank the authors for responding to my comments and improving the paper based on my recommendations.
> >
> > 1. On the second thought and read, I agree with your argument that this indeed could be viewed as external evaluation. However, as mentioned by the authors, please test on other diverse datasets in the future.
> > 2. Thanks for conducting this analysis. The current version gives much more insight into the data efficiency and capability process of the training.
> > 3. I agree that the community needs to work together to release more such datasets publicly.
> > 4. Please include the other downstream tasks into the future extension of this work. I am really looking forward to this.
> > 5. I think the authors should still try to incorporate EndoFM or similar endocscopy related FMs in their work (may be in the future). Complementary way of including them into the training process is quite encouraged.
> > 6&7. Acknowledged.
> >
> > My final question/suggestion is: Do the authors plan to release their dataset or/and the model weights publicly in the future? It would be a great resource for the community working the in the domain of AI for GI domain.

---

> > > ### Author Response · Authors · 2026-01-28
> > >
> > > We thank the reviewer for the encouraging remarks.
> > >
> > > In future work, we plan to further evaluate our approach on additional and more diverse datasets as well as on other downstream tasks. Moreover, we aim to incorporate EndoFM or similar endoscopy-related foundation models into comparative experiments.
> > >
> > > Regarding openness and reproducibility, we have already publicly released the trained model weights at [https://huggingface.co/tofriede/dinov3-upperGI](https://huggingface.co/tofriede/dinov3-upperGI/tree/main). Unfortunately, at this point we are not permitted to publicly release the UpperGI-400K dataset. However, we fully agree that such a release would be highly beneficial for the community, and this is an ongoing topic of internal discussion that we actively aim to address in the future.

---

### Author Rebuttal · Authors · 2026-01-24

**Rebuttal:**

We thank all reviewers for their valuable and constructive feedback. In response to the suggestions raised by two reviewers, we have conducted two additional extensive experiments. First, we analyzed data efficiency by systematically reducing the size of the labeled training set for the esophagitis detection task. Second, we performed a data scaling study to investigate how much in-domain data is required for self-supervised pretraining before diminishing returns are observed. All changes and additions are highlighted in orange in the revised manuscript.

**Supporting Material:**

/attachment/7a0482cb5c93e3f1c727d2828f326f823734eead.pdf

---

### Comment · Area_Chair_LXVz · 2026-01-28

Dear Reviewers,

We are now in the discussion phase. If you have not yet done so, please check the authors’ rebuttal and evaluate how well your concerns have been addressed. I encourage you to engage in discussion with the authors and other reviewers where helpful.

Most importantly, please update your Final Rating after considering the rebuttal and discussion.

Your input is important for a fair meta-review and final decision. Thank you for your continued effort.

AC

---

### Meta-Review · Area_Chair_LXVz · 2026-02-01

**Recommendation:** Accept (Oral)
**Confidence:** 4

**Metareview:**

All three reviewers are positive and agree that this paper is well-motivated and practically valuable for esophagitis detection in upper GI endoscopy. The main strength is a clear and thorough experimental study showing that in-domain self-supervised pretraining on a large unlabeled endoscopy corpus improves downstream performance compared with ImageNet supervised pretraining and natural-image self-supervised pretraining, across multiple public datasets. Reviewers also appreciate the release of code and pretrained weights, which support reproducibility and community impact. In the rebuttal, the authors addressed the remaining concerns by clarifying the evaluation setting and adding/expanding analyses such as data scaling and annotation-efficiency results, which strengthened the empirical evidence and improved presentation. While the methodological novelty is mostly in application and validation rather than introducing a new SSL algorithm, the work is solid, clearly written, and the rebuttal resolves the key issues. Overall, I recommend acceptance.

---

### Decision · Program_Chairs · 2026-02-13

Accept (Poster)